# Biomarker-Development Proteomics in Kidney Transplantation: An Updated Review

**DOI:** 10.3390/ijms24065287

**Published:** 2023-03-09

**Authors:** Vittorio Sirolli, Luca Piscitani, Mario Bonomini

**Affiliations:** 1Nephrology and Dialysis Unit, Department of Medicine, G. d’Annunzio University, Chieti-Pescara, SS. Annunziata Hospital, 66013 Chieti, Italy; 2Nephrology and Dialysis Unit, Department of Medicine, San Salvatore Hospital, 67100 L’Aquila, Italy

**Keywords:** proteomics, kidney transplantation, allograft, rejection, mass spectrometry, biomarker

## Abstract

Kidney transplantation (KT) is the optimal therapeutic strategy for patients with end-stage renal disease. The key to post-transplantation management is careful surveillance of allograft function. Kidney injury may occur from several different causes that require different patient management approaches. However, routine clinical monitoring has several limitations and detects alterations only at a later stage of graft damage. Accurate new noninvasive biomarker molecules are clearly needed for continuous monitoring after KT in the hope that early diagnosis of allograft dysfunction will lead to an improvement in the clinical outcome. The advent of “omics sciences”, and in particular of proteomic technologies, has revolutionized medical research. Proteomic technologies allow us to achieve the identification, quantification, and functional characterization of proteins/peptides in biological samples such as urine or blood through supervised or targeted analysis. Many studies have investigated proteomic techniques as potential molecular markers discriminating among or predicting allograft outcomes. Proteomic studies in KT have explored the whole transplant process: donor, organ procurement, preservation, and posttransplant surgery. The current article reviews the most recent findings on proteomic studies in the setting of renal transplantation in order to better understand the effective potential of this new diagnostic approach.

## 1. Introduction

Kidney transplantation (KT) is, at present, the best choice of therapy for end-stage kidney disease (ESKD) patients, offering better survival, quality of life, and cost-effectiveness than any modality of dialysis [1,2].

Careful surveillance of allograft function is the key to post-transplantation management. Kidney injury may be induced by several different causes that require different approaches for patient management and are indistinguishable by any available noninvasive test. Histologic examination of the kidney allograft through invasive biopsy is currently the gold standard for assessing post-transplant complications [3]. However, allograft biopsy has monitoring drawbacks, is limited by subjective interpretation, is prone to sampling errors, and carries some risk [4].

In clinical practice, serum creatinine, estimated glomerular filtration rate (eGFR), and urinary albumin/protein excretion are the current biomarkers of kidney damage. These parameters are easy to measure, non-invasive, and interpretable, but apply to a later stage of the disease; they are neither sensitive nor specific, and are not connected with the molecular pathophysiology [5]. Thus, there is a need for accurate novel noninvasive biomarkers for continuous monitoring after KT and early diagnosis of allograft dysfunction, which in turn may lead to an improvement in the clinical outcome.

In the field of biomarker discovery, proteins and peptides (the main units of cell function and structure) have great potential, since differences in the composition of proteome and peptidome are indicative of pathologic conditions. The entire set of proteins or peptides expressed in a biological sample (blood, urine, and tissue) can be fully characterized using proteomics or peptidomics, respectively. The analysis of the peptidome and proteome will be treated identically during the remainder of this review.

There are several different techniques that can be used for studies on proteomic and protein biomarkers [5]. Mass spectrometry (MS)-based methods, the most frequently used to retrieve the system proteome, represent a powerful analytical tool for the identification and quantification of a large number of proteins, including posttranslational modifications. These methods include capillary electrophoresis-MS, liquid chromatography-MS (LC-MS), ion trap-MS, matrix-assisted laser desorption/ionization-time of flight MS (MALDI-TOF MS), surface-enhanced laser desorption/ionization-time of flight MS (SELDI—TOF MS), and isobaric tags for relative and absolute quantification (iTRAQ). Gel electrophoresis (GE) methods such as two-dimensional and differential GE are used for the separation of complex protein samples, but have a low detection limit and can only identify a reduced number of proteins. Other proteomic techniques include, among others: protein microarrays, an emerging class capable of high throughput detection for small amounts of sample; Bio-Layer Interferometry, a rapid (1 h) method of detection of biomolecular interactions; enzyme-linked immunosorbent assay (ELISA); and enzyme-linked immunoSPOT (ELIPSOT), a technique resembling ELISA but offering higher sensitivity. Detailed technological features and comparison of proteomic methods are outlined in recent reviews [6,7].

Proteomic technologies allow us to achieve identification, quantification, and functional characterization of proteins/peptides in a sample through a supervised or targeted analysis. Proteomic techniques have been applied extensively these last several years as a tool for biomarker discovery in kidney disease and the management thereof [8,9,10,11,12]. They may yield earlier detection, prognostic assessment, and a prediction of response to treatment, as well as a better understanding of the pathophysiologic mechanisms.

In KT, then, new biomarker molecules are clearly needed for early determination and post-transplant success [5,13,14,15,16]. Ideally, a biomarker should be non-invasive, reliable to monitor, appear early during injury onset, and correlate with the degree of injury [17]. Many studies have investigated proteomic techniques as potential molecular markers discriminating among or predicting allograft outcomes. In the present work, we review the most recent findings on proteomic studies in the setting of kidney transplantation, covering the publication years 2017–2022.

## 2. Clinical Applications of Proteomics in Kidney Transplantation

Proteomic studies in KT have explored the whole transplant process: donor, organ procurement, preservation, and posttransplant surgery. While molecular markers in the donor may be useful in anticipating short- and long-term outcomes, molecular markers after transplantation may improve our knowledge on adaptation and any shortcomings of the graft.

### 2.1. Allograft Quality

The number of patients on waiting lists for a kidney transplant is increasing rapidly. Since organ availability is low, the gap between patients on the list and the number of available kidneys has forced marginal kidneys to be employed from expanded criteria donors (ECD). The outcome of these transplants is, however, generally less favorable than with other donor types, and includes an increased risk of delayed graft function (DGF) and incidences of primary nonfunction (PNF) [18,19]. Innovative methods for the assessment of graft quality are therefore urgently needed [20].

To date, the traditional methods used to assess graft quality have included visual assessment, kidney risk score and the Kidney Donor profile index, histologic analysis after biopsy, microbiologic analysis of storage fluids, perfusion monitoring, and imaging [20].

Proteomics aims to assess biomarkers able to predict graft function in the recipient. Vaughan et al. [21] used LC-MS/MS and immunoblotting to examine the degradation profiles of cytoskeletal proteins in preimplantation biopsies from deceased and living donor kidneys. They provided evidence that, as compared with circulatory death and living donor kidneys (reference cohorts), brain death donor kidneys are more susceptible to the activation of proteolytic processes causing alterations in the podocyte cytoskeleton, predominantly in grafts with suboptimal function (eGFR ≤ 40 mL/min/1.73 m^2^) after 12 months. Interactions between transforming growth factor-beta and calpain-1 may drive the degradation of key cytoskeletal proteins [21].

Efficient preservation of the kidney prior to transplant surgery is a crucial issue in renal transplantation. The functional preservation period is limited to hours, during which proteins, peptides, and other molecules are released into the preservation medium by the organ. Moser et al. [22] compared the levels of different injury biomarkers (neutrophil gelatinase-associated lipocalin (NGAL), metalloproteinase 2, and lactate dehydrogenase) in 41 kidneys undergoing cold perfusion. They found higher levels of injury markers in the perfusate from donors with circulatory death than in those from brain-dead or living donors. Possible byproducts of injury included collagen fragments, immunoglobulin, and albumin [22]. Guzzi et al. [23], in the context of hypothermic machine perfusion, reported a correlation between glutathione transferase levels in the perfusate and acute kidney damage, there being a moderate predictive ability for delayed graft function. More recently, van Leeuwen et al. [18] examined the protein profiles of cadaveric donor kidney perfusate during hypothermic machine perfusion using LC-MS/MS to identify differences between proteomic profiles of kidneys with a good (eGFR ≥ 45 mL/min/1.73 m^2^) and suboptimal outcome 1 year after transplantation. Good outcomes were associated with the upregulation of 22 proteins, including four proteins involved in complement activation of the classical pathway, which suggests that the scavenging by tissue-resident complement proteins detected in perfusate may improve the outcome. On the other hand, 26 proteins proved to be downregulated in good outcome kidneys, among them 14 cytoskeleton proteins [18].

In summary, the results of these studies indicate that proteomic analyses may provide insights into the pathophysiologic mechanisms underlying organ preservation and represent a useful tool to identify and predict graft outcome (Table 1).

Further studies are required to confirm these results. Improving perfusion technologies is also needed to improve organ resistance and the success of transplantation [24]. A recent study profiling kidney proteomes showed that normothermic machine perfusion with urine recirculation allows prolonged preservation of the kidney and ameliorates metabolic processes, rendering the organ more amenable to transplantation [25].

### 2.2. Ischemia-Reperfusion Injury

Ischemia-reperfusion injury (IRI) refers to increased tissue damage after the reperfusion of previously ischemic tissue [26]. In KT, ischemia is inevitable and IRI may cause tissue damage, with complex and still not-well-understood mechanisms. A key role seems to be played by the production of reactive oxygen species (ROS) through anaerobic metabolism in response to hypoxia, pH reduction, and ATP depletion. Reperfusion causes the activation of the immune system with the release of pro-inflammatory cytokines [27,28]. This process can cause PNF or DGF and episodes of acute rejection and transplant fibrosis. Notably, 12 h is the preservation time after which reperfusion induces cell death, with an intensity proportional to the preservation time [29,30].

Lindeman et al. [31] examined the possible metabolic origin of IRI by combining data from sequential arteriovenous graft blood samplings with pre- and post (40 min)-reperfusion tissue biopsies. They found IRI associated with a post-reperfusion metabolic collapse, characterized by the failure of oxidative phosphorylation and activated normoxic glycolysis, causing an inability to sustain the organ’s energy requirements. Their study suggests that efforts to quench IRI should focus on preserving or restoring metabolic competence [31]. In a more recent study, Pasini-Chabot et al. [30] investigated the proteome of kidney endothelial cells (the first cell line impacted by IRI) subjected to cold ischemia for 3–24 h and 6 h of reperfusion. After LC-MS/MS analysis, a heatmap was generated to distinguish protein motifs by their variation, and each group was analyzed using the Cytoscape’s ClueGO application to identify ontology and network associations. Cold ischemia proved to not be a simple slowing down of metabolism since changes in critical pathways were found, including the cytoskeleton structure/transport system, energy metabolism, and gene transcription/translation. Upregulation within these pathways was maintained until 12 h cold ischemia, with the protein expression decreasing thereafter. Upon reperfusion, all expressed proteins were downregulated and no new proteins were detected. This study indicates key proteins for investigation as potential targets for novel strategies to optimize organ quality [30]. These results are summarized in Table 2.

Remote ischemic conditioning (RIC) has been proposed as a therapeutic strategy in the protection of ischemia-reperfusion injury [32,33]. A randomized clinical trial (CONTEXT) evaluating the effects of RIC (preconditioning) compared with non-RIC in 225 transplant recipients showed no clinical improvement of transplant outcomes, however [34]. Analysis by high-resolution tandem mass spectrometry (MS/MS) of samples from CONTEXT recipients showed RIC to be associated with a transient increase in the plasma levels of acute phase response proteins and an accumulation of muscle proteins and abnormal amino acid metabolism in kidney tissue proteomes not reflected in plasma [28]. These results suggest that the RIC regimen applied is not effective and does not elicit a significant molecular response in either the target organ itself or systemically [28].

In summary, a better understanding of the factors potentially leading to IRI may lead to early diagnosis and optimization of treatment. Proteomic identification of biomarkers for IRI may help in this.

### 2.3. Delayed Graft Function

Delayed graft function (DGF) is one of the severe manifestations of IRI in the setting of kidney transplantation and can be defined as the need for dialysis in the first week or weeks after transplantation [35]. The incidence of DGF varies between 19 and 70% in kidney transplants from deceased donors and approximately 10% in transplantation from living donors [36]. It is increased by a cold ischemia time > 24 h, a warm ischemia time > 18 min, high levels of proteinuria and terminal renal function in the donor, dialysis modality before transplantation, or sensitization in re-transplanted patients [37,38]. Clinically, DGF is associated with post-transplant oliguria, prolonged post-operative hospitalization, high rejection rates, and poor graft outcome [36]. Omics sciences including proteomics have been applied in the search for biomarkers associated with DGF in order to identify early preclinical signs of graft dysfunction and improve graft management [36]. Several urinary and blood biomarkers have been proposed. Some studies focused on identifying proteome biomarkers in donor urine. A large (*n* = 469 deceased donors and *n* = 902 corresponding kidney recipients), prospective, observational cohort study by Schroppel et al. [39] investigated the levels of anaphylatoxins C3a and C5a in donor urine at the time of organ procurement. The complement system is activated by ischemia and increases inflammatory response and subsequent renal damage through the release of pro-inflammatory cytokines [40]. In their cohort, Schroppel et al. found a three-fold increase in C5a concentration in the urine of donors with stage two and three acute kidney injury (AKI) as compared with donors without AKI. DGF in the cohort was 32% and proved to be positively correlated with donor C5a. In adjusted analyses, donor C5a was significantly associated with an elevated risk of DGF in donor urine without AKI [39]. A recent phase I/III double blind placebo-controlled study examined the ability of C1 esterase inhibitor (C1INH) to prevent IRI/DGF in kidney transplant patients at risk for DGF [41]. Patients received either C1INH (*n* = 35) or placebo (*n* = 35) intraoperatively and at 24 h. Though the primary end point (need for hemodialysis (HD) in the first week after transplantation) was not met, treatment with C1INH significantly reduced the need for HD sessions 2 to 4 weeks posttransplant and improved the long-term graft function [41].

Mansour et al. [42] evaluated whether the concentration of monocyte chemoattractant protein-1 (MCP-1) in urine from deceased donors is associated with graft outcomes. MCP-1 is a chemokine produced by many cell types that is involved in inflammation and repair after renal injury. Donor urinary MCP-1 showed minimal clinical utility, as it was not independently associated with DGF nor was it predictive of recipient graft function at 2 years [42]. More effectively, Braun et al. [43] collected small urinary extracellular vesicles (suEVs) from 22 living kidney donors and recipients. Using unbiased proteomic analysis, they showed temporal patterns of suEV protein signature and cellular processes which are involved in early response and long-term graft adaptation. The activation of complement was particularly prominent. The study also sought to identify potential prognostic markers of future allograft function. Abundance of phosphoenol pyruvate carboxylase (PCK2) in the suEV proteome 24 h after transplantation proved to be predictive for kidney function one year after the transplant, as validated in an independent cohort of 22 additional kidney recipients. The study suggests the potential of suEV as biomarkers, though the small number of patients requires confirmation in larger studies [43].

Other proteomic analyses were applied to urine samples from transplant recipients to identify protein biomarkers of DGF. Williams et al. [44], using Targeted Urine Proteome Assay (TUPA), identified a panel of the top four protein biomarkers of DGF. The panel, including the C4b-binding protein alpha chain, guanylin, serum amyloid P-component, and immunoglobulin superfamily number eight, showed an area under the curve (AUC) of 0.891, a sensitivity of 77.4%, and a specificity of 82.6% [44]. Bank et al. [45] examined the urinary tissue inhibitor of metalloproteinases-2 (TIMP-2) and insulin-like growth factor binding protein-7 (IGFBP7) in 74 recipients from cardiac death donors. TIMP-2 and IGFBP7 are validated predictive biomarkers of AKI [46]. In Bank et al.’s study, higher urine TIMP-2 levels adequately identified patients with DGF (AUC 0.89) and prolonged DGF (AUC 0.77), while IGFBP7 did not. Correcting TIMP-2 for urine osmolality (TIMP-2/mOsm) enhanced the predictability of DGF, and consecutive TIMP-2/mOsm values showed a reduction in TIMP-2/mOsm before an increase in eGFR. These results indicate TIMP-2 as a promising biomarker for the prediction and the duration of DGF in individual patients [45]. Finally, though urine NGAL is of potential usefulness, it is worth emphasizing that is has low specificity and, thus, its clinical application is limited and its results are inconclusive [47].

Studies investigating protein biomarkers of DGF have also been performed in blood. In 27 cases of deceased donor transplantation (DGF *n* = 11), Hu et al. [48] showed reduced plasma levels of corin, a serum protease which generates atrial natriuretic peptide, at 24 h after surgery in patients presenting DGF after IRI. In a murine model of renal IR injury, the protein level of Corin was found to be downregulated [48].

The discovery of novel non-invasive biomarkers of DGF is of critical importance if we are to improve the quality of graft management. The current evidence (Table 3) strongly suggests that proteomic biomarkers may be predictive of DGF before and immediately after renal transplantation, making for prompt therapeutic intervention. The proteins identified may be candidates for further validation studies.

### 2.4. Allograft Rejection

Rejection refers to an attack on the graft by the host immune apparatus. Despite improvements to immunosuppressive agents that have significantly reduced rejection events, kidney transplant patients are still at risk of allograft rejection (AR) [49]. Immune graft rejection can be clinically classified as hyperacute, acute, and chronic rejection [50]. Hyperacute rejection is now quite rare and is caused by antibodies pre-existing in the patient’s bloodstream against antigens on the allograft; it occurs within a few minutes to a few hours after surgery. Episodes of AR are most prevalent in the first weeks after transplantation [51] and can be categorized into T-cell-mediated rejection (TCMR) and antibody-mediated rejection (ABMR). In the first case, the underlying pathophysiologic mechanism is lymphocyte proliferation, which induces cytotoxic effects on the renal tubular epithelial cells, along with tubulitis and invasion of the arteries by mononuclear cells, causing arteritis and sometimes necrosis of the blood vessels. ABMR, by contrast, can be seen within the first year post-transplant and is mediated by donor-specific antibodies that target human leukocyte antigen (HLA) or non-HLA antigens on the donor endothelium. This interaction produces cellular cytotoxicity and complement activation with endothelial cell injury secondary to the recruitment of leukocytes and platelet aggregation. Chronic rejection (CR) is the damage resulting from the residue of antigraft antibodies or lymphocyte, causing failure to control rejection by immunosuppressive therapy. CR is discussed in the section Chronic Allograft Dysfunction.

Histologic examination of kidney allograft is the gold standard to detect rejection, though there are the previously mentioned concerns about kidney allograft biopsy. The use of proteomics provides a unique opportunity to gain insights into the mechanisms involved in KT rejection processes. Several molecules in blood and urine have been evaluated as reflecting the molecular process in the allograft and, hence, may be predictive biomarkers for early diagnosis and monitoring. Most recent results on proteomics studies are reported below.

#### 2.4.1. Acute Rejection

Various biomarkers have been noticed in the urine of KTRs with AR. These include cytokines, extracellular matrix proteins, and renal tubular cell constituents, such as NGAL, kidney injury molecule-1 (KIM-1), interleukin (IL)-1R, IL-20, IL-18, C-X-C motif chemokine 9 (CXCL9), and C-X-C motif chemokine 10 (CXCL10). Shahbaz et al. [52] found that a biomarker panel consisting of KIM-1 identified AR from non-AR with an AUC of nearly one, there being an upregulation of this molecule in patients who underwent AR. Moreover, beginning within the first two weeks post-transplant, an increase in the soluble cluster of differentiation thirty (CD30), which mediates the poise between T helper type one and type two immune responses, predicted AR with a sensitivity of 88.8% [53].

Using a MS technology based on iTRAQ, Zhang et al. [54] detected 109 proteins that differentially expressed the AR group and controls. Several proteins were upregulated including properdin, keratin 1, lipoprotein a, and vitamin D-binding protein, which may have a role in the pathogenesis of AR. In particular, high levels of properdin, the only known positive regulator of complement activation playing an important role during early renal ischemia-reperfusion injury, may be significantly involved in the development of AR, and anti-properdin therapy may be beneficial in this pathology [54]. Mertens and coworkers [55] performed a multicenter case–control study to recognize a urinary biomarker for ABMR alternative to biopsy. The primary endpoint of the study was the diagnostic accuracy of the urinary biomarker for ABMR. They identified a set of 10 urinary proteins (alpha-1 B glycoprotein, afamin, apolipoprotein A1, A4, Ig heavy constant alpha1, gamma 4, leucine rich alpha2 glycoprotein 1, alpha-1 anti-trypsin (SERPINA1), antithrombin, and transferrin) that discriminated patients with (*n* = 60) and without (*n* = 189) ABMR with the same accuracy as histological examination. The negative predictive value of the 10-protein marker set for the exclusion of ABMR was 0.99 and the diagnostic accuracy was independent of the reason for performing the biopsy, as well as the time after transplantation, and proved better than the accuracy of gross proteinuria (AUC = 0.76) [55]. In a subsequent study in 36 KT recipients (KTRs), examination of the urinary proteome succeeded for the first time in identifying urinary epidermal growth factor (EGF) as a possible marker for early diagnosis of AR [56].

The role of proteomics in differentiating ABMR, acute cellular rejection (ACR), or acute tubular necrosis (ATN) in kidney biopsies from transplant recipients is suggested by a recent study [57]. A total of 107 glomerular and 112 tubulointerstitial proteins proved to be significantly differentially expressed in ABMR versus ACR, and many others were dysregulated in ABMR as compared with ATN. Glomerular and tubulointerstitial expression of laminin subunit gamma-1 (LAMC1) were decreased in AMR, as were glomerular nephrin (NPHS-1) and receptor-type tyrosine-phosphatase O. Moreover, proteomic analysis revealed up-stimulation of galectin-1, an immunomodulatory protein linked to the extracellular matrix, in ABMR glomeruli. Conversely, anti-HLA class I antibodies increased cathepsin-V (CTSV) expression in human glomerular endothelial cells. More recently, Chaveau et al. [58] used laser microdissection of glomeruli from formalin-fixed graft biopsies combined with MS to investigate the proteome modification of 11 active and 10 chronic active ABMR cases compared with eight controls with stable graft. Among the over 1300 proteins detected, 77 proved to be deregulated in glomerulitis (the histological hallmark of active ABMR); three proteins extracted from this protein profile, guanylate-binding protein-1 (GBP1), targeting thymidine phosphorylase (TYMP), and WARS1, displayed marked overexpression in glomerular endothelial cells, indicating endothelial stress during active ABMR. In chronic active ABMR (transplant glomerulopathy), 137 proteins were deregulated; the most relevant pathways were indicative of complement-mediated mechanisms, wound healing processes through coagulation activation, and ultimately reorganization and expansion of the extracellular matrix [58]. Overrepresentation of extracellular matrix-regulator proteins suggests potential targets of therapeutic interest for preventing extracellular matrix expansion.

However, noninvasive evaluation of AR in renal transplant patients using urinary proteomics may fail to prevent premature graft failure. In a prospective, multicenter, phase III diagnostic study enrolling 624 KTRs from different European countries, the use of a 14-peptide panel (previously validated in a small cohort) on urine samples taken immediately before graft biopsy diagnosing TCMR did not predict borderline TCMR and was poorly predictive of acute TCMR, as the sensitivity of the model was 0.66, specificity 0.47, and AUC 0.60 [59]. These results very likely reflect the limitations associated with the population used to build the predictive model [50].

A further aid to the early diagnosis of problems related to transplant organ dysfunctions has in recent years been ascribed to extracellular vesicles (EVs). Three types of EVs (exosomes, microvesicles, and apoptotic bodies) have been discriminated on the basis of their biogenesis [60]. EVs have progressively emerged as being biologically active and a source of biomarkers for various diseases [61].

Exosomes (EX) are cell-derived membrane vesicles surrounded by lipid bilayers present in fluid such as blood and urine, which have a significant role in cellular activities, intercellular communication, and waste management [62]. Several proteomics studies have revealed the composition of urinary EX secreted from different nephron segments and claimed their relevance to the renal pathophysiology of kidney disease [63], while evidence is accumulating that exosome contents are also involved in the rejection of transplantation. The presence of intragraft infiltration of T cells is one of the hallmarks of the diagnosis of acute cellular rejection after KT, and hence a higher level of CD3-positive urinary EX in patients with AR could reflect T-cell infiltration [64]. A subsequent proteomics evaluation revealed elevated levels of urinary exosome tetraspanin-1 and hemopexin in subjects with TCMR as compared with subjects without rejection [65], while EX mRNA transcripts (CCl4, gp130, CAV1, TNF alpha, SH2D1B, and atypical chemokine receptor 1) were helpful in distinguishing antibody-mediated rejection patients from cellular rejection recipients [66]. Kim et al. [67] examined the potential use of urinary exosomal proteins as biomarkers for ABMR in 36 kidney transplant patients. Out of 1820 exosomal proteins in the discovery cohort, four were specifically associated with ABMR: cystatin C (CST3), serum paraoxonase/arylesterase 1, retinol-binding protein 4, and lipopolysaccharide-binding protein [67]. Al-Nedawi et al. [68] reported the potential of blood microvesicles as a novel tool for predicting the outcome of KT. They used proteomics to study the protein content of circulating microvesicles from KTRs with poor (*n* = 10) or better (*n* = 10) transplant outcomes according to eGFR and from age-matched healthy controls. A series of well-defined protein clusters were highlighted per category of subject evaluated, by which it was possible to differentiate KTRs from healthy subjects and distinguish between transplanted patients according to their eGFR. Proteomic analysis of blood microvesicles could thus help to discriminate between transplant recipients with different graft prognoses [68]. It is important to note in this connection that Castellani et al. [69], working with heart transplant recipients, observed a significantly higher number of plasma EVs in patients with antibody- or cellular-mediated rejection processes than in transplanted patients without rejection processes.

Results of recent proteomic studies on AR are summarized in Table 4.

The development of proteomic technologies has opened up new opportunities for non-invasive diagnosis of AR, whether cell-mediated or antibody-mediated, thereby hastening and improving the management of kidney transplantation. Current evidence indicates potential roles for proteomic biomarkers in predicting or diagnosing AR (Table 4). These promising biomarkers may enable patient risk stratification, monitoring over the entire post-transplant course, and improvements in therapeutic decision-making. Their clinical utility needs confirmation in future studies.

#### 2.4.2. Chronic Allograft Dysfunction

Chronic allograft dysfunction (CAD) encompasses different etiologies of chronic dysfunction including true CR, drugs, and viruses [16]. Biomarkers able to discriminate between true CR and either AR or other causes of CAD are important, since treatments differ.

By proteomic profiling using MALDI-TOF MS and magnetic beads, Hussien et al. [70] studied 75 subjects divided into three groups, equally distributed (*n* = 25 per group): Group one, patients with chronic allograft nephropathy (CAN); Group two, stable allograft function; and Group three, healthy subjects. Five peaks differentiated CAN patients from the control group with a sensitivity and specific rate of 100%, and the five peaks also distinguished between transplant patients with normal kidney function and the control groups, with the sensitivity rate being little less than 97% and a specific rate of 95.5% [70]. In a cross-sectional multicenter study, Jung et al. [71] enrolled 26 patients with biopsy-proven chronic active antibody-mediated rejection (CAMR), 57 with long-term graft survival, and 10 patients rejection-free. Proteomic analysis of urinary EVs identified six proteins in CAMR patients (apolipoprotein A1, transthyretin, polymeric immunoglobulin receptor, hemopexin, zinc-alpha-2 glycoprotein, and ceruloplasmin) as being expressed differently (to a significant degree) from the long-term graft survival group. Zinc-alpha-2 glycoprotein (AZGP1) displayed potential as a specific proteomic biomarker for CAMR [71]. The results of these studies are summarized in Table 5.

The most complete and exhaustive data about biomarkers found in CR have been obtained with the online informatics tool STRING [72]. The STRING database collects and scores evidence from a variety of input sources, aiming to integrate both physical interactions and functional associations between proteins, resulting in comprehensive protein networks. The most pertinent pathways of proteins identified in CR prove related to immune response (13 proteins), defense response (11 proteins), response to alcohol and response to molecules of bacterial origin (seven proteins for both), and regulation of intracellular transport (eight proteins) [5]. Moreover, the use of STRING in comparing biomarkers upregulated in AR and CR disclosed that inflammation was absent in the list of CR, which suggests that CR is a more mature noninflammatory immune response than the inflammatory process of AR [5].

Altogether, these studies mark a noteworthy effort, in a noninvasive way, to detect CAD in an early phase and to differentiate true CR from other forms of chronic graft nephropathies, which may have therapeutic implications.

### 2.5. BK Virus Infection

Infections in KT are common and represent an important cause of curtailed life [73,74]. They can be caused by either common or opportunistic pathogens and are mainly due to induction and maintenance immunosuppression.

BK virus (BKV) is one of the most common pathogens in post-transplant infections. BKV is a double-stranded DNA member of the Polyomaviridae family; it induces a common viral infection in children without residual complications and remains latent in the renal tubular epithelium of healthy subjects. After transplantation, BKV may reactivate, leading to viruria in 30–40% of patients and to BKV viremia in 10–20% [75]. A proportion of these patients will develop BK-virus-associated nephropathy (BKVAN), and up to 90% of them may lose their graft within a year [75,76]. BKV viruria results from the lysis of infected cells, which enables the virus to escape into the tubular lumen. Denudation of the urogenital basement membrane, which occurs in the presence of high levels of BKV viruria, causes vascular spread and subsequent BKV viremia. BK viruria precedes viremia (by about 4 weeks), just as viremia precedes BKVAN [76]. Polymerase chain reaction (PCR) is a useful non-invasive test to identify BKV viruria (>7 log_10_ copies/mL) in urine and viremia (>4 log_10_ copies/mL) in plasma for concomitant BKVAN with a sensitivity of 100% and specificity of 78 and 92%, respectively. The Society of Transplantation Infectious Disease Community of Practice 2019 (AST-IDCOP) recommends this type of screening [77].

The results of recent proteomic studies on BKV infection in kidney transplant patients are highlighted in Table 6.

Caller et al. [78] used tandem mass tags (TMT) and MS3 mass spectrometry technology to analyze BKV-induced changes in protein expression among renal epithelial cells. Interestingly, most of the functional clusters identified as upregulated in BKV infection related to cell cycle activity and regulation associated with the G2 and M phases. The same study also showed an alteration to the MDM2 (reduced) and p53 (increased) axis. Thus, infected cells do not progress to the mitotic phase [78]. The BKV-induced changes in protein expression of the host cell are not excessive, showing that there is an adaptation between this virus and humans.

A recent study by Wang et al. [79] used proteomics to analyze the plasma protein repertoire in patients with BKV-negative to BKV-activated transition. Twelve differentially expressed proteins were identified, with S100A8 and S100A9 being the top two upregulated proteins in patients with BK infection. S100A8/A9, known as calprotectin, has an antibacterial role, but if it increases it may induce tissue injury and aggravate organ dysfunction [80]. S100A8/A9 proved to be more highly expressed in patients with BK viremia than in patients positive for urinary BK. Furthermore, in patients with BK viremia, plasma S100A8/9 protein was an independent risk factor for allograft function impairment [79].

Bruschi et al. [76] performed a proteomic analysis of the protein content of urinary extracellular vesicles (microvesicles and exosomes) to better define the biological mechanisms associated with BKV infection. The analyses demonstrated an amplification of several biological processes including immunity, complement activation, epithelial to mesenchymal transition, and renal fibrosis in extracellular urinary vesicles from KTRs with BKV infection as compared with controls without BKV. Thirty-six proteins discriminated the two study groups, and among them only two proteins were over-expressed in the urine of transplanted patients with BKV infection: deoxyribonuclease 2 alpha (DNASE2) and biphosphate 3′-nucleotidase 1 (BPNT1). BPNT1 has been associated with abnormal uromodulin levels, which in turn correlates with the formation of polyomavirus-haufen in the kidney [76]. The upregulation of DNASE2 could target viral DNA for degradation [81]. Note that the proteomic profile of patients with BKV viruria was comparable to that of transplanted patients with both BKV viruria and viremia. This study, though performed in a small cohort of patients, comprises a further attempt to obtain mechanistic insights into BKV infection in KTR. It is suggested that due to the activation of virus-related biological mechanisms, preventive therapeutic strategies (mainly reduced immunosuppression) should be considered, even in KTRs with BKV viruria alone [76]. A recent study showed that KTRs with high levels of circulating BKV viremia exhibited significantly reduced T-cell reactivity, supporting the link between immunosuppression and BKV reactivation [75].

Histologically, BKV-induced kidney damage can easily be confused with T-cell-mediated rejection (TCMR), causing diagnostic delays and therapeutic errors. Although blood and urine proteomics techniques are revealing possible predictive biomarkers of BKV infection, one recent work performing proteomics analysis on paraffin-embedded samples of renal tissue from transplanted patients quantified 2798 proteins, of which 638 were associated with TCMR, and 740 were significantly altered in BKV compared with STA samples [82].

**Table 6 ijms-24-05287-t006:** Recent proteomic studies on BKV infection in the setting of kidney transplantation.

Application	Type of Sample	Main Findings	Ref.
Analysis of BKV-induced changes in protein expression	Renal epithelial cells	Infected cells do not progress to the mitotic phase	Caller et al. [78], 2019
Analysis of plasma protein repertoire in transplanted patients with BKV-negative to BKV-activated transition	Plasma	Among 12 differentially expressed proteins, S100A8 and S100A9 were the top upregulated proteins In patients with BKV viremia, S100A8/A9 was an independent risk factor for graft function impairment	Wang et al. [79], 2022
Analysis of the protein content of urinary EVs in patients with BKV infection compared with controls without BKV	Urine	36 proteins discriminated the two groups. Two proteins (DNASE2 and BPNT1) alone were over-expressed in transplanted patients with BKV infection Proteome profiles in BKV viruria were comparable to those of patients with both BKV viruria and viremia	Bruschi et al. [76], 2022
Assessment of proteomic signature to differentiate TCMR and BKV nephropathy from STA	Kidney biopsies	Out of 2798 quantified proteins, 638 associated with TCMR and 740 associated with BKV significantly altered compared with STA	Song et al. [82], 2020

EVs, extracellular vesicles; DNASE2, deoxyribonuclease 2 alpha; BPNT1, biphosphate 3’-nucleotidase 1; TCMR, T-cell-mediated rejection; STA, stable kidney tissue.

In summary, BKV is a serious infection in kidney transplant recipients; we need early biomarkers predicting the infection as a warning to implement preventive therapy, primarily based on reducing immunosuppressive therapy. Recent research has shown that proteome profiling can identify novel biomarkers able to discriminate BKV infection/nephropathy and to provide new mechanistic insights into BK infection. Additional studies need to be conducted to confirm these results in larger cohorts of patients.

### 2.6. Other Investigations

Other proteomic studies in the setting of kidney transplantation are reported in Table 7.

Some proteomic studies addressed the potential toxic effects of immunosuppressive therapy in transplant patients. A study by Jacobs-Cachá et al. [83] suggested fascin-1, an actin-binding protein, as a putative urinary biomarker to assess damage caused by calcineurin inhibitors (CNI) in the kidney tubular compartment. Conceivably, urine fascin-1 might result from EX released by tubular cells [83]. Recently, Carreras-Pianella et al. [84] used urinary EVs to investigate the nephrotoxic effects of CNIs and, specifically, their chronic toxicity (CNIT), which can lead to renal fibrosis. The authors enrolled patients treated with CNI who had normal kidney function, suffered from CNIT, or presented interstitial fibrosis and tubular atrophy (IFTA). They observed that the proteome of CNIT was significantly enriched in gene sets related to epithelial cell differentiation, probably because of the death of tubular epithelial cells that forces them to regenerate. A higher expression of cell-linker proteins from the uroplakin and plakin families was also found in CNIT than in IFTA, suggesting a toxic effect by CNI on the urothelium [84]. These results corroborate the significant roles of EX as a source of pathogenic molecules and non-invasive biomarkers in KT. Using them during management of immunosuppressive treatment may help clinicians to preserve and perhaps prolong the allograft survival and function.

Among the general population, progressive chronic kidney disease (CKD) is perceived in its advanced stages by a continuous decline in glomerular filtration. A classifier based on 273 urinary peptides, termed CKD 273, is well suited to the early detection of CKD and prognosis of progression [85]. Recently, 52 living donor kidney recipients with a long-term follow up were urine sampled at month 24 after transplantation and analyzed for the patients’ peptide profiles using the CKD 273 classifier, which showed a significant positive correlation with serum creatinine at every time point [86]. Using the composite endpoint graft loss and death within the next six years from proteomic evaluation as a classification criterion, receiver operating characteristics (ROC) analysis revealed an AUC for CKD273 of 0.89, while the stratification of patients revealed a hazard ratio of 16.5 with 95% for the prevalence of graft loss in the case of CKD273 positivity [86].

Finally, in a recent study Jalal et al. [87] used an aptamer-based assay to analyze the proteome of plasma and circulating EVs in KTRs, and from it identified biomarkers of vascular injury and inflammation-cardiovascular disease (CVD), which are still the most common causes of mortality in kidney transplant patients [88]. As compared with healthy controls, the plasma levels of angiogenesis proteins proved significantly increased in KTRs [87]. Interestingly, some of these proteins correlated with urinary albumin excretion, a biomarker of CVD. The top pathways activated included Ephrin receptor signaling, transforming growth factor-beta, and serine biosynthesis. Furthermore, EV proteome analysis showed a prominent pro-inflammatory profile. The study was the first to indicate that pathways of angiogenesis and inflammation, activated in KT, might represent potential therapeutic targets [87].

**Table 7 ijms-24-05287-t007:** Miscellaneous recent proteomic studies in the setting of kidney transplantation.

Application	Type of Sample	Main Findings	Ref.
Assessment of nephrotic effects of CNIs	Urine	Fascin-1 as a putative urinary biomarker to assess kidney damage by CNIs	Jacobs-Cachà et al. [83], 2017
Assessment by EVs of nephrotoxic effects of CNIs in patients using CNI with normal graft function or suffering from CNI toxicity	Urine	Proteome of patients with CNI toxicity enriched in gene sets related to epithelial cell differentiation and in cell-linker proteins from uroplakin and plakin families	Carreras-Pianella et al. [84], 2020
Proteome analysis of plasma and circulating EVs to identify biomarkers of vascular injury and inflammation	Plasma	Activation of pathways of angiogenesis and inflammation, which could represent potential therapeutic targets	Jalal et al. [87], 2021

CNI, calcineurin inhibitor; EVs, extracellular vesicles.

## 3. Conclusions

Prompt recognition and treatment of adverse events occurring during the lifetime of a transplanted kidney is of major importance to save the allograft and, often, the patient. In KT there is an unmet need for reliable early biomarkers distinguishing between the different forms of graft damage so that an appropriate and effective therapy may be prescribed. We have here outlined how results obtained from proteomic studies show the potential value of such omics methodology in improving transplant outcomes. Proteomic research in KT has revealed its potential for developing valuable tools that improve patient management by enabling the prediction, early diagnosis, prognostic assessment, and therapeutic monitoring of pathological events related to kidney allografts. These results may be further supported by larger current clinical trials, which are studying proteomic biomarkers in KT. The “Mass Spectrometry-based Proteomics in Microvascular Inflammation Diagnosis in Kidney Transplantation. (TranSpec)” study is a single group diagnostic trial aiming to assess the sensitivity and specificity of MS-based proteomics for diagnosing microvascular inflammation through biopsy and urine samples (ClinicalTrials.gov identifier NCT04851145). The “Proteogenomic Monitoring and Assessment of Kidney Transplant Recipients (Mini-Kidney)” trial is a prospective, observational single-center monitoring study that seeks to validate proteogenomic panels for AR and CAN/interstitial fibrosis and tubular atrophy in blood, urine, and graft tissue of kidney transplant recipients scheduled for standard of care biopsies (NCT01531257). A further ongoing trial is the “Molecular Biological and Moleculargenetic Monitoring of Therapy After Kidney Transplantation (MoMoTxRes)” study (NCT01515605). This is a prospective observational cohort study which analyzes over a 10-year period whether noninvasive diagnostic molecular monitoring may improve the outcome of KT. Finally, proteomics is employed in the exploratory efficacy of the study “FIH (First in Human) Trial Evaluating Safety of TUM012 to Minimize Ischemic Reperfusion Injury in Kidney Transplantation” (NCT05246618). It is the first in-human, randomized, double-blind, placebo-controlled trial which examines the safety and tolerability of ex vivo infusion of TUM012 to reduce IRI.

However, to translate proteomic biomarkers from bench to bedside is a challenge. One major problem is that only a few transplant studies range beyond the discovery phase of biomarker development to include fully independent validation studies demonstrating the clinical utility of the biomarkers proposed [89]. Most trials are also of limited size. Adequately powered clinical studies including larger patient cohorts are required to statistically identify robust clinical biomarkers. The increased availability of high-definition equipment such as MS in hospital biochemistry services is also of major importance in favoring the introduction of proteomic-based tools in clinical practice [13]. Finally, collaboration between investigators involving data sharing and the realization of a central proteome database is also a critical goal. A comprehensive urine proteome database was recently generated from a variety of urine samples, including KTRs with AR or stable graft [90]. The database may serve as a reference resource for facilitating the discovery of potential urinary proteomic biomarkers.

The results of all of such efforts will be to clarify the role of proteomics in the management of kidney transplant patients.

## Figures and Tables

**Table 1 ijms-24-05287-t001:** Recent proteomic studies on graft quality in the setting of kidney transplantation.

Application	Type of Sample	Main Findings	Ref.
Assessment of degradation profiles of cytoskeletal proteins	Preimplantation biopsy from deceased and living donors	Compared with cardiac death and living donor kidneys, brain death donor kidneys showed activation of proteolytic processes causing alterations in the podocyte cytoskeleton	Vaughan et al. [21], 2022
Analysis of levels of biomarker injury (NGAL, metalloproteinase 2, lactate dehydrogenase)	Perfusate	Highest perfusate concentrations of biomarker injuries in kidneys from deceased cardiac death donors, followed by brain death donors and living donor allografts	Moser et al. [22], 2017
Assessment of the protein profile of deceased donor kidney perfusate that predicts a good and suboptimal graft outcome	Perfusate	Upregulation of 22 proteins and downregulation of 26 proteins associated with a good graft outcome 1 year after transplant	Van Leeuwen et al. [18], 2021

NGAL, neutrophil gelatinase-associated lipocalin.

**Table 2 ijms-24-05287-t002:** Recent proteomic studies on ischemia-reperfusion injury in the setting of kidney transplantation.

Application	Type of Sample	Main Findings	Ref.
Assessment of the possible metabolic origin of ischemia-reperfusion injury	Tissue biopsies, plasma	Metabolic collapse post-reperfusion characterized by failure of oxidative phosphorylation and activated normoxic glycolysis with inability to sustain the organ’s energy requirements	Lindeman et al. [31], 2020
Assessment of the proteome of kidney endothelial cells subjected to cold ischemia	Tissue (kidney endothelial cells)	Upregulation until 12 h cold ischemia of critical pathways including the cytoskeleton structure/transport system, energy metabolism, and gene transcription/translation	Pasini-Chabot et al. [30], 2021

**Table 3 ijms-24-05287-t003:** Recent proteomic studies on delayed graft function in the setting of kidney transplantation.

Application	Type of Sample	Main Findings	Ref.
Assessment of C3a and C5a as biomarkers for graft outcome	Donor urine	In donor urine without AKI, C5a associated with elevated risk of recipient DGF	Schroppel et al. [39], 2019
Assessment of urinary MCP-1 with graft outcome	Donor urine	Low clinical utility of donor urine MCP-1 due to lack of correlation with DGF and graft function	Mansour et al. [42], 2017
Analysis of the proteome of suEVs from living kidney donors and recipients	Urine	Temporal patterns of suEV protein signature and cellular processes involved in early and long-term graft Abundance of PCK2 may have predictive value for graft function	Braun et al. [43], 2020
TUPA to identify biomarkers of delayed recovery after kidney transplantation	Urine	A panel including C4b-binding protein alpha chain, guanylin, serum amyloid P component, and immunoglobulin superfamily number 8 distinguished DGF	Williams et al. [44], 2017
Evaluation of urinary changes of TIMP-2 and IGFBP7 for diagnostic utility for predicting DGF	Urine	Urinary TIMP-2, but not IGFBP7, as potential biomarker for the occurrence and duration of DGF	Bank et al. [45], 2019

MCP-1, monocyte chemoattractant protein-1; DGF, delayed graft function; AKI, acute kidney injury; suEVs, small urinary extracellular vesicles, PCK2, phosphoenol pyruvate carboxylase; TUPA, Targeted Urine Proteome Assay; TIMP-2, tissue inhibitor of metalloproteinases-2; IGFB7, insulin-like growth factor binding protein-7.

**Table 4 ijms-24-05287-t004:** Recent proteomic studies on acute rejection in the setting of kidney transplantation.

Application	Type of Sample	Main Findings	Ref.
Analysis of differentially expressed proteins in AR and controls	Serum	Upregulation in AR of properdin, keratin 1, lipoprotein(a), and vitamin D-binding protein	Zhang et al. [54], 2020
Searching for urinary biomarkers to recognize ABMR, alternative to biopsy	Urine	A set of 10 urinary proteins discriminated patients with and without ABMR with the same accuracy as histologic examination	Mertens et al. [55], 2020
Examination of urinary proteome as an early marker of AR	Urine	Identification of EGF as a possible marker of early AR	Heidari et al. [56], 2021
Differentiating ABMR, ACR, or acute tubular necrosis	Kidney biopsy	A total of 107 glomerular and 112 tubulointerstitial proteins proved significantly differentially expressed in ABMR vs. ACR, and many others in ABMR vs. acute tubular necrosis	Clotet-Freixas et al. [57], 2020
Analysis of proteome modification of active and chronic active ABMR compared with stable graft	Kidney biopsy	77 proteins deregulated in ABMR and 137 proteins deregulated in chronic active ABMR	Chauveau et al. [58], 2022
Evaluation of urinary exosome biomarkers of TCMR	Urine	Elevated levels of urinary exosomes tetraspanin-1 and hemopexin in patients with TCMR	Lim et al. [65], 2018
Evaluation of urinary exosomal proteins as biomarkers of ABMR	Urine	Four proteins (cystatin C, serum paraoxonase/arylesterase 1, retinol-binding protein 4, and lipopolysaccharide-binding protein) specifically associated with ABMR	Kim et al. [67], 2022

AR, acute rejection; ABMR, antibody-mediated rejection; EGF, epidermal growth factor; ACR, acute cellular rejection; TCMR, T-cell-mediated rejection.

**Table 5 ijms-24-05287-t005:** Recent proteomic studies on chronic allograft dysfunction in the setting of kidney transplantation.

Application	Type of Sample	Main Findings	Ref.
Identification of urinary profile of patients with CAN	Urine	Five peaks distinguished CAN	Hussien et al. [70], 2020
Assessment of the proteome of urinary EVs in biopsy-proven chronic active ABMR compared with patients with long-term graft survival and patients rejection free	Urine	Six proteins (apolipoprotein A1, transthyretin, polymeric immunoglobulin receptor, hemopexin, zinc-alpha-2 glycoprotein, and ceruloplasmin) identified patients with chronic active ABMR Zinc-alpha-2 glycoprotein displayed potential as a specific biomarker for chronic active ABMR	Jung et al. [71], 2020

CAN, chronic allograft nephropathy; EVs, extracellular vesicles; ABMR, antibody-mediated rejection.

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
