# Peer review of "Biomarker-Development Proteomics in Kidney Transplantation: An Updated Review"

_ijms, 2023, doi:10.3390/ijms24065287_

Round 1

Reviewer 1 Report

The review of recent findings on proteomic studies in the context of kidney transplantation may be of interest to readers in the proteomics and transplantation fields, but the writing should be improved. Specific areas for improvement include:

1.    The title should be more engaging to capture the reader's attention.

2.    The organization of the review is difficult to understand and should be reorganized. For instance, “IRI in allograft characteristics”?

3.    The focus on "recent articles" (2017-2022) should be consistent throughout the review and not include older articles.

4.    There are a few paragraphs/sentences which are almost the same as published papers, the authors should rephrase sentences from other literature.

e.g.

line 44:

Proteins and peptides are the main functional and structural units of the cell and hence of great biomarker potential, since qualitative and quantitative differences in the proteome and peptidome composition reflect pathologic conditions.

line 392:

"The STRING database obtains protein–protein interactions from experimental data, databases, coexpression analysis, detection of shared selective signals, across genomes, text searches of existing articles, and transference of protein interactions."

5.    The introduction should provide more details on proteomics techniques.

6.    The table should be rearranged to place the categories in the first column and summarize per category.

7.    The authors should proofread for typos and errors, such as the one on line 463-464 [(> 1 × 107 copies/mL) in urine and viremia (> 1 × 104 copies/mL)].

Author Response

We thank the Reviewer for her/his evaluation of our review.

Response to the comments of Reviewer (page and line when indicated refers to the revised manuscript with revisions highlighted).

  1. The title should be more engaging to capture the reader's attention.

The title has been modified as follows: Biomarker – Development Proteomics in Kidney Transplantation: An Updated Review.

  1. The organization of the review is difficult to understand and should be reorganized. For instance, “IRI in allograft characteristics”?

According to the suggestion of Reviewer, we have somewhat modified the organization of the review. In particular, previous section 2.1 Allograft characteristics has been changed, the revised version containing 3 different sections (2.1 Allograft characteristics; 2.2 IRI; 2.3 DGF). Title of section 2.5 has been modified from Infections into BK virus Infection.

  1. The focus on "recent articles" (2017-2022) should be consistent throughout the review and not include older articles.

The revised version of the review includes articles published in the period 2017-2022 only. Previous reference #4 and #36 have been replaced by more recent ones, the other “old” references have been removed.

  1. There are a few paragraphs/sentences which are almost the same as published papers, the authors should rephrase sentences from other literature.

We thank the Reviewer for the comment and have modified the paragraphs sentences:

line 44:

Proteins and peptides are the main functional and structural units of the cell and hence of great biomarker potential, since qualitative and quantitative differences in the proteome and peptidome composition reflect pathologic conditions.

“In the field of biomarker discovery, proteins and peptides (the main units of cell function and structure) have great potential, since differences in the composition of proteome and peptidome are indicative of pathologic conditions.” (line 46)

line 392:

"The STRING database obtains protein–protein interactions from experimental data, databases, coexpression analysis, detection of shared selective signals, across genomes, text searches of existing articles, and transference of protein interactions."

“The STRING database collects and scores evidence from a variety of input sources, aiming to integrate both physical interactions and functional associations between proteins, resulting in comprehensive protein networks.” (line 460)

  1. The introduction should provide more details on proteomics techniques.

The introduction now contains more details on proteomics techniques as follows:

“There are several different techniques which can be used for studies on proteomic and protein biomarkers (5). Mass spectrometry (MS)-based methods, the most frequently used to retrieve the system proteome, represent a powerful analytical tool for identification and quantification of a large number of proteins including posttranslational modifications. These methods include capillary electrophoresis-MS, liquid chromatography-MS, ion trap-MS, matrix-assisted laser desorption/ionization-time of flight MS (MALDI-TOF MS), surface-enhanced laser desorption/ionization-time of flight (SELDI—TOF MS), and isobaric tags for relative and absolute quantification (iTRAQ). Gel electrophoresis methods such as two-dimensional and differential GE are used for separation of complex protein samples but have a low detection limit and can only identify a reduced number of proteins. Other proteomic techniques include among others: protein microarrays, an emerging class capable of high throughput detection for small amounts of sample; Bio-Layer Interferometry, a rapid (1 hour) method of detection of biomolecular interactions; enzyme-linked immunosorbent assay (ELISA); and ELIPSOT, a technique resembling ELISA but offering higher sensitivity. Detailed technological features and comparison of proteomic methods are outlined in recent reviews (6, 7).” (lines 53-68)    

  1. The table should be rearranged to place the categories in the first column and summarize per category.

The table has been modified according to the comments of Reviewer 2. The original table has been broken up into smaller tables (n=7) corresponding to each topic of the review. Column category has been removed from the table since each category is now indicated in the table legend. 

  1. The authors should proofread for typos and errors, such as the one on line 490-491 [(> 1 × 107 copies/mL) in urine and viremia (> 1 × 104 copies/mL)].

We apologize for errors, which have been corrected:

 >7 log10 copies/mL replaces >1 x 107 copies/mL in urine

 >4 log10 copies/mL replaces >1 x 104 copies/mL (viremia)

Author Response

We thank the Reviewer for her/his appreciation of our review and for the comments.

Response to the comments of Reviewer (page and line when indicated refers to the revised manuscript with revisions highlighted):

  • However, I would suggest the author perhaps summarize it in a way of figure. This will enhance understanding of the readers to this complex topic.

We thank the reviewer for the suggestion. However, we feel that breaking up the table into smaller tables (see comment below) may enhance understanding of the readers to this topic.

  • I would also recommend break up the table 1 into smaller table corresponding to each topic of KT.

According to the suggestion of the Reviewer, we have broken up the original single table into smaller tables, corresponding to each topic of KT discussed in the manuscript. The revised version contains 7 tables.

  • More importantly, I would prefer the author to summary what are the current evidence in each section.

The current evidence in each section has been summarized.

  • Lastly, it is undeniable that we are needing the larger studies to confirm these findings. I would suggest the authors add these ongoing studies into the conclusion perhaps current clinical trial that are studying these markers.

The Conclusion section now contains a paragraph (lines 617-635) reporting ongoing clinical studies on the use of proteomics in KT.

Minor suggestion.

  1. Use the term end-stage kidney disease rather than end-stage renal disease

The term end-stage kidney disease is used (page 1 line 29)

  1. Use the term eGFR rather than GFR on page 1 line 37

eGFR has been used (page 1 line 39)

  1. Page 2, line 88: what does MS stand for?

MS stands for Mass Spectrometry, as now indicated in the Introduction (page 2 line 54)

  1. Page 2, line 89: what is the control group of study by Vaughn et al. This should be mentioned in  the text.

The control group of study by Vaughan et al. has been mentioned: “They provided evidence that, as compared to circulatory death and living donor kidneys (reference cohorts)”  (page 3 line 113-115)

  1. Page 3, line 141: IRI instead of IR

 IRI has replaced IR (page 5 line 171)

  1. Page 4, line 163: I disagree with this statemen. “Delayed graft function (DGF) is the manifestation of IRI in the setting of kidney trans- 163 plantation”. Etiology of DGF varies and IRI  is one of them as the author stated later in the paragraph. Would consider changing to DGS is one of the severe manifestations of IRI.

According to the comment of the Reviewer, we have modified the statement as follows: “Delayed  graft function (DGF) is one of the severe manifestations of IRI…..” (page 5 line 201)

  1. Page 4 line 176: , observational cohort study “by” Schrappel et al

“by” Schroppel et al. (page 6 line 215)

  1. Page 4 line 189: reduce the need for HD at what time?

The need for HD sessions was reduced 2 to 4 weeks posttransplant. This information has been added    to the manuscript (page 6 line 227-228)

  1. Page 4 line 192: MCP-1; please add hyphen

The hyphen for MCP—1 has been added (page 6 line 230)

  1. Page 6 line 273: Ramirez-Sandoval and coworkers observed that high urinary or serum levels of NGAL were associated with reduced graft survival at 1 year [57]. Is this study focusing on acute rejection, if not, I would remove from this section.

Reference by Ramirez-Sandoval and coworkers has been removed.

  1. Page 10 line 473: Make sure the number is correct for BK viral load.

We apologize for errors, which have been corrected (page 13  line 490-491):

 >7 log10 copies/mL replaces >1 x 107 copies/mL in urine

 >4 log10 copies/mL replaces >1 x 104 copies/mL (viremia)

Reviewer 3 Report

This paper is very well-written.  It is valuable to accept this paper.

Author Response

We wish to thank the Reviewer for her/his appreciation of our manuscript.

Round 2

Reviewer 1 Report

The author has taken my concern into account and addressed it in the latest version of the manuscript. Please review again the punctuation carefully to ensure it is correct (such as line 514).

Author Response

We thank the Reviewer for appreciation of the changes we made to the manuscript.

Regarding punctuation, as outlined by the Reviewer at line 514 (antibac-terial), we believe this is due to the hyphenation sofware used by the journal. Another example is at line 567 (cal-cineurin). We tried to correct but it was not possible due to the sofware settings. However, we noticed that in a clean version of the manuscript antibacterial is correctly reported.